# Compound Multi-branch Feature Fusion for Real Image Restoration

## Abstract

Image restoration is a challenging and ill-posed problem which also has been a long-standing issue. However, most of learning based restoration methods are proposed to target one degradation type which means they are lack of generalization. In this paper, we proposed a multi-branch restoration model inspired from the Human Visual System (i.e., Retinal Ganglion Cells) which can achieve multiple restoration tasks in a general framework. The experiments show that the proposed multi-branch architecture, called CMFNet, has competitive performance results on four datasets, including image dehazing, deraindrop, and deblurring, which are very common applications for autonomous cars. The source code and pretrained models of three restoration tasks are available at `https://github.com/publish_after_accepting/CMFNet`.

## 1 Introduction

Image restoration is a low-level vision task, and is usually the preprocessing step to improve the performance of the high-level vision task, such as image classification, object detection or image segmentation. As its name implies, Image Restoration is meant to restore an image free from degradation. Common degraded types include additive noise, blur, and JPEG blocking effect. However, with the rapid development of computer vision in recent years, the types of degradation that image restoration can handle are becoming more and more diverse, like super-resolution (Ledig et al., 2017; Zhang et al., 2018c), single image dehazing (Zhang & Patel, 2018; Qin et al., 2020), image deraining (Ren et al., 2019; Zhang et al., 2019a) or even demoireing (Yuan et al., 2019; 2020).

For decades, traditional image restoration was mostly based on external image prior (He et al., 2010; Shi et al., 2013) or dictionary learning (Hu et al., 2010; Dong et al., 2011) which are generally called model-based algorithms. Although these methods performed acceptably on the ill-posed problem as above mentioned, there are some shortages for model-based optimization methods, like time-consuming, computationally expensive and difficult to restore complex image textures. With the success of learning-based algorithms on both high-level (Krizhevsky et al., 2012) and low-level vision tasks, convolution neural network (CNN) is dominant in the computer vision field nowadays.

CNN-based methods not only outperform the model-based algorithms, but also have the state-of-the-art results in lots of image restoration tasks. Numerous networks and techniques of learning-based algorithms, including encoder-decoders (Mao et al., 2016), generative models (Ledig et al., 2017), residual learning (Zhang et al., 2017), and attention mechanisms (Chen et al., 2016) have been proposed to enhance the performance of image restoration tasks. In addition to different network architectures, the complex blocks (e.g., residual block, residual dense block (RDB) (Zhang et al., 2018c)) or modules (Convolution Block Attention Module (CBAM) (Woo et al., 2018) and Dual Attention Units (DAU) (Zamir et al., 2020a;b)) are presented to replace the naive convolutions.

We observed that most of restoration models only focus on one degradation type to restore (Nah et al., 2017b; Zhang et al., 2018c; Ren et al., 2019; Chen et al., 2020; Qin et al., 2020). The performances of these restoration models are impressive, but are not able to restore other degradation tasks. To put it another way, the generalization ability of these models has to be improved. Fortunately, all of the aforementioned restoration tasks have common final objective (i.e., converting an image from illegible and low-quality to satisfactory and high-quality). Thus, we proposed a general framework inspired from the Human Visual System (HVS) which could conduct the restoration tasks, including deblurring, dehazing and deraining. The reason why we focus on these three restoration tasks is

because they usually occur at the environment of autonomous cars. Hence, the proposed framework is practical and prospective in the future applications.

The proposed compound multi-branch feature fusion image restoration architecture, called CMFNet has several important characteristics. First, inspired from the early path of the HVS, we use three types of existing attention blocks, including channel attention, spatial attention, and pixel attention blocks to simulate the Retina Ganglion Cells (RGCs) independently. The main idea is to separate different attention features from stacking multiple complicated blocks to multiple branches with simple block architecture. Second, we use the Supervised Attention Module (SAM) proposed by Waqas Zamir et al. (2021) to improve the performance. We also remove the supervised loss between the output images from SAM and ground truth images, because we think it would limit the learning of network. Thirdly, we propose a mixed skip connection (MSC) to replace traditional residual connection with a learnable constant, which makes the residual learning more flexible on different restoration tasks. Finally, we optimize our CMFNet end-to-end by a novel loss function, which provides more selections about the loss function rather than the common losses. All of these key components about our architecture will be described in Section 3 in detail.

Overall, our main contributions of this paper can be summarized as the following:

- We propose a restoration model CMFNet inspired by the RGCs which can handle multiple image restoration tasks, including image deblurring, dehazing and deraindrop with single framework.

- We propose the skip connection with a learnable parameter named mixed residual module to integrate the branch information. The experiments proved that it is better than the monotonous residual pathway aggregation.

- A new loss function which comprises two widely used evaluation metrics (i.e., Peak Signal-to-Noise Ratio (PSNR) and Structure Similarity (SSIM) Index) in restoration task is proposed.

- We demonstrate the competitive results of our CMFNet in both synthetic and real-world datasets for three types of restoration tasks, especially the improvements of SSIM values. In other words, the performances of our model are closer to the human sense.

## 2 RELATED WORK

With the rapid development of hardware (e.g., Graphics Processing Unit (GPU)), the deep learning methods become more and more mature, which make the computer vision prosperously grow in recent years. However, as aforementioned, most of methods perform well in specific restoration task. In this paper, we propose a general architecture to solve deblurring, dehazing and deraining. Also, we are going to introduce the related works about RGCs and attention mechanism. Then, we will describe the representative methods for each task in detail.

**Retina Ganglion Cells (RGCs).** Human is the final receiver of the images. The pictures we saw were passed from the complicated visual neural network, and the early receivers from the real world are RGCs, which is mainly composed of 3 types of cells: Parvocellular cells (P-cells), Koniocellular cells (K-cells) and Magnocellular cells (M-cells). They are sensitive to different external stimuli. First, P-cells are sensitive to the shape and color of images, and account for 80% of the RGCs. The second one is the K-cells, the smallest RGCs, as small as dust, which mainly respond to changes in color. The third one is M-cells, the biggest RGCs, and it only transmits the light/dark signal, and is more sensitive at low spatial frequencies than high spatial frequencies. All of these cells have independent pathways toward the Lateral Geniculate Nucleus (LGN). Some computer vision researches such as Image Quality Assessment (IQA) (Chang et al., 2020) and Image Reconstruction (Zhang et al., 2020) are based on RGCs, too.

**Attention Mechanism.** Due to the success of the application on the high-level vision tasks, like object detection and image classification, attention mechanism can also be used in low-level vision tasks (Anwar & Barnes, 2019). Well known attention modules or blocks are mainly distinguished by the dimension of the generated mask, such as channel attention (Zhang et al., 2018b), spatial

attention, pixel attention (Chen et al., 2016) and combined attention blocks (Qin et al., 2020; Zamir et al., 2020b). Although the attention could enhance the performance in general, some researchers consider that attention will increase the inference and training time of the model (Liu et al., 2020), and the benefit of using it is very small.

**Image Dehazing.** Image dehazing has the conventional model-based methods. For example, some researchers (Narasimhan & Nayar, 2000; 2002) provided an important model to approximate the haze effect which is shown as: $\hat{x} = x \odot m + \boldsymbol{A} \odot (1 - m)$, where $\hat{x}$ and $x$ mean the degradative images and restored images, respectively. $\boldsymbol{A}$ and $m$ are the global atmospheric light and medium transmission map, and $\odot$ represented as pixel-wise multiplication. In addition, He et al. (2010) proposed an image-prior method depends on the statistical results called Deep Channel Prior (DCP) and has the promising results. As for learning-based methods, the techniques such as attention, feature fusion (Qin et al., 2020) and contrastive learning (Wu et al., 2021) are widely used to ameliorate the single image dehaze performance. Moreover, they outperform the traditional prior-based image dehazing.

**Image Deraining.** For raining dataset, it is almost impossible to get the real-world rainy images until now, so most of experiments are based on synthetic images. And previous tradition deraining methods (Barnum et al., 2010; Bossu et al., 2011) almost all use linear-mapping transformations to restore the rainy images. However, these methods are not robust when the rainy input has variations. Thus, the learning-based model is here again (Zhang et al., 2019a; Chen et al., 2021)! But we want to emphasize all of these methods are trained on artificial rainy images. On the other hand, the deraindrop restoration task which is the branch of image deraining can acquire the real-world dataset. Qian et al. (2018) used two pieces of exactly the same glasses to obtain the real-world raindrop dataset. Compared with the synthesized rain streaks dataset, the Raindrop dataset is closer to the images captured from cameras.

**Image Deblurring.** Because the real-world supervised blur data is difficult to acquire, most of traditional deblur methods, like deconvolution (Krahmer et al., 2006; Carasso, 2001), image prior (Joshi et al., 2010; Shi et al., 2013) are generally tested on synthesized images from $\hat{x}$ to $x$, which can be represented as: $\hat{x} = x \otimes k + n$, where $\hat{x}$ is the blurred image generated from clean image $x$, $k$ is the blur kernel or called convolution kernel, $\otimes$ denotes the convolution operator and $n$ is additive noise. However, the model-based methods are not good at dealing with real-world blur such as hand-shake motion blur or defocus blur. On the contrary, CNN-based methods can handle real-world blur very well if we have the dataset of pair images (Nah et al., 2017b). Tao et al. (2018) proposed a multi-scale method based on encoder-decoder recurrent network (SRN) which is the first method integrating the Recursive Neural Networks (RNN) into deblur model. Some methods (Kupyn et al., 2018; 2019) based on Generative Adversarial Network (GAN) also get the competitive evaluation scores on real-world deblur results. Recently, multi-stage architecture network (Waqas Zamir et al., 2021; Chen et al., 2021) have state-of-the-art results in the deblur restoration task.

## 3 PROPOSED METHOD

In this section, we mainly introduce the multi-branch network CMFNet, and provide more explanations about the components we proposed in the following subsections.

### 3.1 CMFNET

The architecture of the proposed compound multi-branch feature fusion network (CMFNet) is illustrated in Fig. 1. CMFNet consists of three branches which simulate the P-cells, M-cells and K-cells after receiving signals from cone and rod cells. Each branch is based on the U-Net (Ronneberger et al., 2015), and we use different attention blocks to replace naive convolutions in each branch. The output features of encoder-decoder branch networks will enter the Residual Attention Module (RAM) which is obtained by removing the loss calculation from (Waqas Zamir et al., 2021). And we replace $1 \times 1$ convolutions in the original SAM with $3 \times 3$ convolutions (Chen et al., 2021). Then, each RAM will generate two outputs (e.g., $F_C$ and $I_C$ in the upper branch as shown in Fig. 1), where $F_C \in \mathbb{R}^{H \times W \times C}$ denotes the feature map generated by the mask (M) obtained by passing the image $I_C$ through sigmoid, and $I_C \in \mathbb{R}^{H \times W \times 3}$ denotes the output image obtained by passing the degraded image through $3 \times 3$ convolution. The $I_C$, $I_P$ and $I_S$ from three branches will be

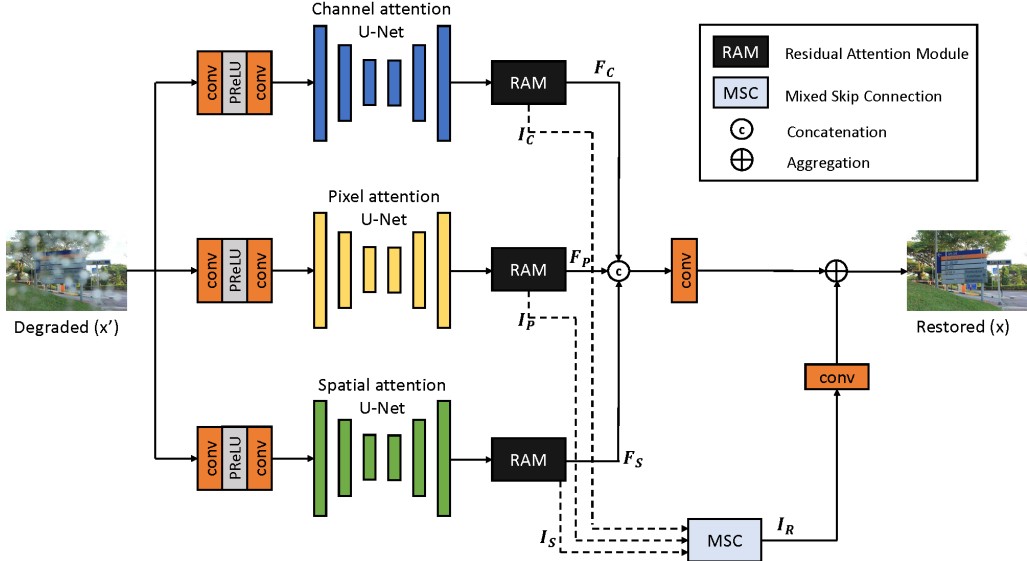

Figure 1: Proposed compound multi-branch feature fusion network (CMFNet) architecture. Each branch is based on 3 layers of U-Net with 3 attention blocks in each scale.

fed into the Mixed Skip Connection (MSC) to obtain the residual image $I_R$. Finally, $I_R$ and the concatenated fusion features ($F_C$, $F_P$, and $F_S$) will both go through $3 \times 3$ convolution to aggregate the restored image.

## 3.2 BRANCH NETWORK

The CMFNet comprises three branch networks which are based on the U-Nets, and each of branch network extracts shallow features by two $3 \times 3$ convolutions and one Parametric ReLU (PReLU). All of the branch networks have 3 layers of U-Net architecture with 3 blocks at each scale as shown in Fig. 2. The main difference among the three branch networks is the type of attention blocks used in the U-Net. We will introduce each branch network in detail as below:

**P-cells network.** We use the pixel attention block (PAB) (Zhao et al., 2020) to simulate the P-cells, which are sensitive to the color and shape (edge) of images. The PAB is shown in Fig. 4(b), where it generates the 3-D attention mask ($M \in \mathbb{R}^{H \times W \times C}$) without any pooling or sampling, which means the output feature map has local information. However, because the dimension of the mask is the largest among the three attention blocks, the computation time is the most. On the contrary, the PAB can acquire the most accurate attention features. We observed that the PAB and P-cells have one-to-one correspondence: 1) the transmission speed of P-cells is the slowest since the pixel attention map has the most parameters, and 2) the spatial resolution of P-cells is the biggest which is related to the largest dimension of the pixel attention mask.

**M-cells network.** As for M-cells, it is almost exactly the opposite of the P-cells. M-cells have the fastest transmission speed and the smallest spatial resolution, which can be explained by the channel attention block (CAB). The CAB (Fig. 4(c)) is proposed by (Zhang et al., 2018b) and is widely used on a lot of restoration tasks (Zamir et al., 2020b; Zhao et al., 2020; Waqas Zamir et al., 2021). It uses global average pooling (GAP) to squeeze the input features from 3-dimension to 1-dimension, and then generates the 1-D attention mask ($M \in \mathbb{R}^C$). Because of the GAP, the output features have non-local (global) information, corresponding to the response to the luminance of images for the M-cells.

**K-cells network.** In either transmission speed or spatial resolution, K-cells are in the medium among the three. We simulate it by the spatial attention block (SAB) illustrated in Fig. 4(d) which can generate the 2-D attention mask ($M \in \mathbb{R}^{H \times W}$). The same as channel attention, spatial attention is also operated by the GAP or max average pooling (MAP) to squeeze features to 2-dimension.

Because of the pooling, the output features have non-local (global) information. In other words, the K-cells mainly responds to changes in color which also belongs to the global information.

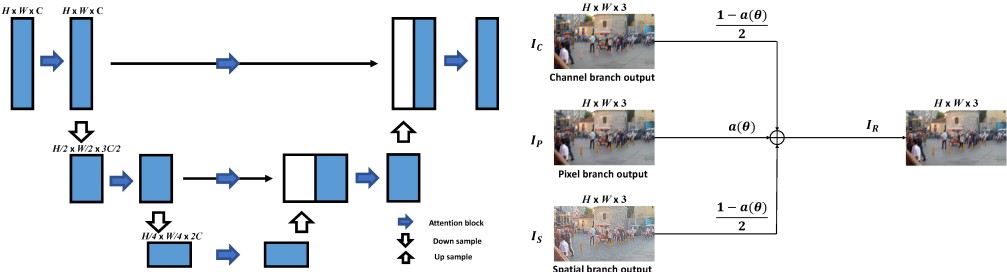

Figure 2: Branch network architecture      Figure 3: Mixed Skip Connection (MSC).

## 3.3 MIXED SKIP CONNECTION

Residual connection was first proposed by (He et al., 2016) in image recognition tasks. There are countless deep learning models using it nowadays. In our CMFNet, we proposed a mixed skip connection (MSC) which integrates the output images from RAM of each branch. Fig. 3 is the framework of MSC, which has three input pathways and uses a learnable constant $a(\theta)$ to optimize the whole architecture by the backward propagation. The main pathway (output image of Pixel branch network) multiplies by the parameter $a(\theta)$, and the others (output images of both Channel and Spatial branch networks) multiply by $(1 - a(\theta))/2$. As a result, the output residual image $I_R$ from MSC can be represented as:

$$I_R = MSC(I_C, I_P, I_S) = (\frac{1 - a(\theta)}{2}) * I_C + a(\theta) * I_P + (\frac{1 - a(\theta)}{2}) * I_S, \quad (1)$$

where $I_C$, $I_P$ and $I_S$ are the output images from channel, pixel and spatial branch networks, respectively. $a(\theta)$ is a learnable parameter from sigmoid activation function which is bounded from 0 to 1 by parameter $\theta$. We can also think of it as a weighted average skip connection, and we will prove the proposed MSC is effective in enhancing the performance by the ablation study in Section 4.6.

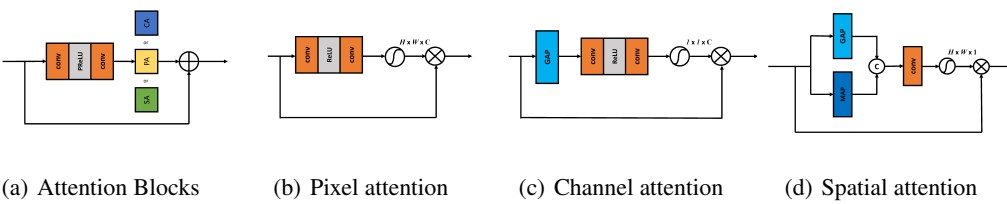

(a) Attention Blocks     (b) Pixel attention     (c) Channel attention     (d) Spatial attention

Figure 4: Attention blocks framework (a) and different attentions modules (b), (c) and (d).

## 3.4 LOSS FUNCTION

In the image restoration task, we usually used the Mean Absolute Error (MAE) or Mean Square Error (MSE) as a loss function to optimize the network. However, both L1 and L2 losses belong to pixel loss which did not consider the global information, so it is usually suffered from the over-smoothing problem. There were many loss functions proposed, such as perceptual loss or even compound loss that combines several loss functions together to solve the problem. To focus on the tradeoff between human sense and metric scores, we propose the new loss function to optimize our CMFNet end-to-end as shown below:

$$\mathcal{L}_{total} = \mathcal{L}_{ps}(X, Y) + \alpha \mathcal{L}_{edge}(X, Y), \quad (2)$$

where $X \in \mathbb{R}^{B \times C \times H \times W}$ denotes the degradation images, $B$ is the batch size of training data, $C$ is the number of feature channels, $H$ and $W$ are the size of images. And $Y$ represents the ground-truth

data. The total loss function is composed of two terms, $\mathcal{L}_{ps}$ and $\mathcal{L}_{edge}$. The $\mathcal{L}_{ps}$ is the loss which is comprised of Peak Signal-to-Noise Ratio (PSNR) and Structure Similarity (SSIM) Index as follows:

$$\mathcal{L}_{ps} = \frac{1 - SSIM(X,Y)}{PSNR(X,Y) + \omega}, \tag{3}$$

where the parameter $\omega$ is a small constant which is empirically set to 0.005. It is used to stabilize the value which can prevent the occurrence of Not a Number (NaN) or infinity when PSNR is extremely small. Our proposed PS loss has two advantages. First, it is suitable for different image restoration tasks because it adopts two standard metrics as the loss function. The other advantage is PS loss does not need additional parameters compared to simply combining different loss functions in several terms with weighting parameters. In other words, it will save time in finding the optimal parameter values by performing many experiments. The second term $\mathcal{L}_{edge}$ is the edge loss which refers to (Jiang et al., 2020) and can be represented as:

$$\mathcal{L}_{edge} = \sqrt{||\Delta(X) - \Delta(Y)||^2 + \varepsilon^2}, \tag{4}$$

where $\Delta$ means the Laplacian operator. The constants $\alpha$ in Eq. (2) and $\varepsilon$ in Eq. (4) are set to 0.05 and $10^{-3}$, the same as (Waqas Zamir et al., 2021).

## 4 EXPERIMENTS

### 4.1 EXPERIMENT SETUP

**Implementation Details.** Our CMFNet is an end-to-end trainable model without any pretrained networks and implemented by PyTorch 1.8.0 with one NVIDIA GTX 1080Ti GPU. We train separate models for three different tasks and the feature channel numbers are all set to 96. The models are trained using Adam optimizer with initial learning rate of $2 \times 10^{-4}$, and decreased to $1 \times 10^{-6}$ by cosine annealing strategy. Because of the limitation of the hardware equipment, we train our network on $256 \times 256$ patches with a batch size of only 2 for $4 \times 10^4$ iterations. Random flip and rotation are applied as data augmentation.

**Evaluation Metrics.** As mentioned in Section 3.4, PSNR is widely used in evaluating the quality of restored images. However, it is sometimes not close to real human perception when PSNR metric is high. Therefore, we also consider the SSIM index in our experiments, which can represent human perceptual feelings more accurately. Note that both PSNR and SSIM values are all the higher the better, and the unit of PSNR is decibel (dB). The training datasets for each restoration task are described as follows.

### 4.2 EXPERIMENT DATASETS

**Image Dehazing.** Since we want our proposed CMFNet could handle the real-world haze, we use the I-Haze (Ancuti et al., 2018a) and O-Haze (Ancuti et al., 2018b) which contain 30 and 45 real-world high definition hazy pair images for training, respectively. And we test the proposed CMFNet on real-world haze dataset D-haze (Ancuti et al., 2019) that contains 55 pair images with the resolution of $1600 \times 1200$.

**Image Deraindrop.** We use the DeRainDrop dataset (Qian et al., 2018) for training and testing. It provides 861 image pairs for training and has two testing datasets (i.e., testA and testB). TestA is a subset of testB, which contains 58 pairs of good aligned images. TestB has 249 image pairs with a small portion of images which are not perfectly aligned.

**Image Deblurring.** For image deblurring, we train the images on the synthesized GoPro dataset (Nah et al., 2017a) as most deblur methods Waqas Zamir et al. (2021); Chen et al. (2021) did. GoPro dataset has 3,124 pair blurred images with the size of $1280 \times 720$, including 2,013 images for training and 1,111 blurred/sharp images for testing. As in Waqas Zamir et al. (2021), we also use the model pretrained on GoPro to test the HIDE (Shen et al., 2019) dataset which contains 2,025 test images by human-aware motion blur.

## 4.3 Image Dehazing Performance

For image dehazing task, we compare the proposed method with the prior-based method (He et al., 2010), learning-based methods (Cai et al., 2016; Li et al., 2017; Liu et al., 2019a; Hong et al., 2020; Qin et al., 2020; Dong et al., 2020), and the contrastive learning method (Wu et al., 2021). Table 1 shows that our method achieves the best SSIM score (0.533) on D-Haze dataset (Ancuti et al., 2019), which means our restored images are closer to human perception. It should be noted that other proposed dehazing methods could not restore the real-world hazy images very well as shown in Fig. 5.

Table 1: Image Dehazing results on Dense-Haze (D-Haze) dataset (Ancuti et al., 2019). Best and second best scores are **highlighted** and underline, respectively.

Table 2: Image Deraindrop results on DeRain-Drop test dataset (Qian et al., 2018). Best and second best scores are **highlighted** and underlined, respectively.

| D-Haze (Ancuti et al., 2019) | | | DeRainDrop (Qian et al., 2018) | | |
|---|---|---|---|---|---|
| Method | PSNR | SSIM | Method | PSNR | SSIM |
| DCP (He et al., 2010) | 10.06 | 0.386 | Eigen13 (Eigen et al., 2013) | 23.74 | 0.799 |
| AOD-Net (Li et al., 2017) | 13.14 | 0.414 | Pix2Pix (Isola et al., 2017) | 27.73 | 0.876 |
| GridDehazeNet (Liu et al., 2019a) | 13.31 | 0.368 | DeRaindrop (w/o GAN) | 29.25 | 0.785 |
| DehazeNet (Cai et al., 2016) | 13.84 | 0.425 | D-DAM (Zhang et al., 2021) | 30.63 | 0.927 |
| KDDN (Hong et al., 2020) | 14.28 | 0.407 | $A^2$Net (Lin et al., 2020) | 30.79 | 0.926 |
| FFA-Net (Qin et al., 2020) | 14.39 | 0.452 | BPP (Michelini et al., 2021) | 30.85 | 0.918 |
| MSBDN (Dong et al., 2020) | 15.37 | 0.486 | DuRN (Liu et al., 2019b) | 31.24 | 0.926 |
| AECR-Net (Wu et al., 2021) | **15.80** | 0.466 | DeRaindrop (Qian et al., 2018) | **31.57** | 0.902 |
| **CMFNet (Ours)** | 14.46 | **0.533** | **CMFNet (Ours)** | 31.49 | **0.933** |

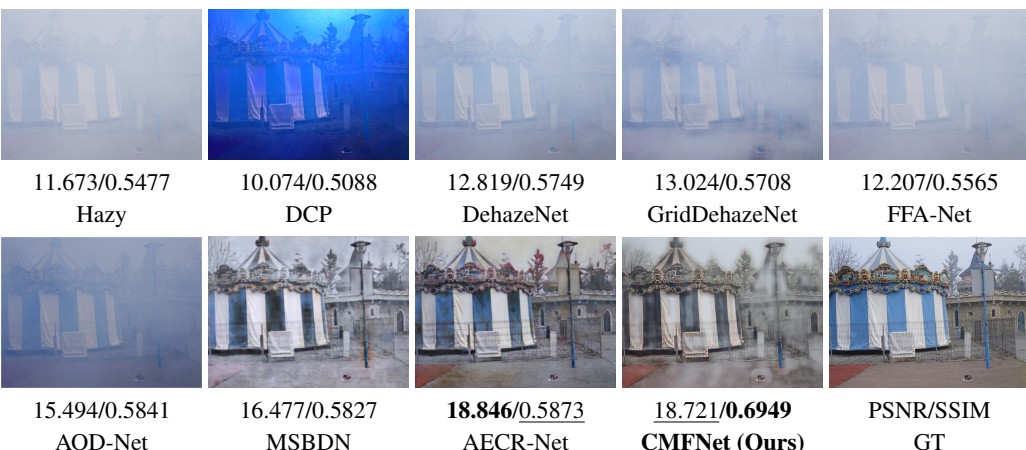

| | | | | |
|---|---|---|---|---|
| 11.673/0.5477 | 10.074/0.5088 | 12.819/0.5749 | 13.024/0.5708 | 12.207/0.5565 |
| Hazy | DCP | DehazeNet | GridDehazeNet | FFA-Net |
| 15.494/0.5841 | 16.477/0.5827 | **18.846**/0.5873 | 18.721/**0.6949** | PSNR/SSIM |
| AOD-Net | MSBDN | AECR-Net | **CMFNet (Ours)** | GT |

Figure 5: Visual comparisons for image dehazing on the D-Haze dataset (Ancuti et al., 2019) .

## 4.4 Image Deraining Performance

In Table 2, we report the PSNR/SSIM scores of deraindrop task compared with several deraindrop methods on the DeRainDrop testA dataset. We follow Qian et al. (2018)'s quantitative evaluation which transforms the testing images from RGB to YCbCr color space to calculate PSNR and SSIM metrics. Our CMFNet achieves the best SSIM score (0.933) and second best PSNR score (31.49 dB). Fig. 6 illustrates visual results on DeRainDrop testB images. We are not able to generate the deraindrop images of both BPP(Michelini et al., 2021) and D-DAM (Zhang et al., 2021) methods because the source codes are not available yet when we submit our paper. However, it still shows

our method effectively removes raindrops and restored images are visually closer to the ground-truth than the other models.

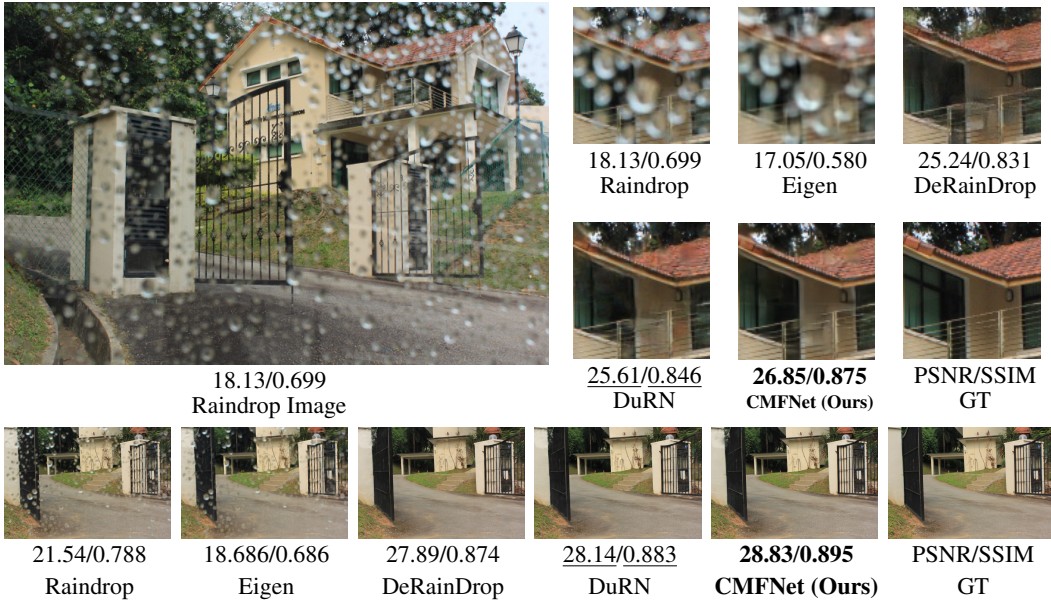

Figure 6: Visual comparisons for image deraindrop on the DeRainDrop dataset (Qian et al., 2018) .

## 4.5 IMAGE DEBLURRING PERFORMANCE

As for deblurring, Table 3 shows the results with both synthetic and real-world dataset. In Table 3, although our CMFNet does not achieve the best performance, the evaluation scores are still acceptable (ranked in the middle). It means the proposed model could handle the degradations occurred on the images taken by cameras of autonomous cars. Due to page limit, visual comparison of deblurring results are provided in the appendix.

Table 3: Image Deblurring results on GoPro (Nah et al., 2017a), HIDE (Shen et al., 2019) datasets.

| Methods | GoPro (Nah et al., 2017a) | | HIDE (Shen et al., 2019) | |
| | PSNR | SSIM | PSNR | SSIM |
| --- | --- | --- | --- | --- |
| Xu et al. (2013) | 21.00 | 0.741 | - | - |
| DeblurGAN(Kupyn et al., 2018) | 28.70 | 0.858 | 24.51 | 0.871 |
| Nah et al. (2017a) | 29.08 | 0.914 | 25.73 | 0.874 |
| DeblurGAN-v2 (Kupyn et al., 2019) | 29.55 | 0.934 | 26.61 | 0.875 |
| Zhang et al. (2018a) | 29.19 | 0.931 | - | - |
| SRN (Tao et al., 2018) | 30.26 | 0.934 | 28.36 | 0.915 |
| DMPHN (Zhang et al., 2019b) | _31.20_ | _0.940_ | _29.09_ | _0.924_ |
| MPRNet (Waqas Zamir et al., 2021) | **32.66** | **0.959** | **30.96** | **0.939** |
| **CMFNet (Ours)** | 30.54 | 0.913 | 28.23 | 0.882 |

## 4.6 ABLATION STUDY

To demonstrate the contribution of each component of the proposed CMFNet, we present ablation studies to analyze different elements, including multi-branch, proposed PS loss and MSC. The ablation studies are experimented on the GoPro dataset (Nah et al., 2017a) with training image patches of size $128 \times 128$ for $2 \times 10^4$ iterations. The whole ablation study results are shown in Table 4.

**Number of branches.** We set our model from single branch to triple branch framework, which proves that the multi-branch architecture is effective in improving the performance (from 28.00 dB to 28.88 dB).

**Attention blocks of branch networks.** Because each U-Net branch network is comprised of different attention blocks to simulate the RGCs, we present the experiments of different attention blocks, and validate the pixel attention block has better performance (28.63 dB) than channel and spatial attention blocks (28.00 dB and 28.26 dB, respectively).

**Choices of loss functions.** As for the choice of the loss functions, we provide the results of widely used MAE (L1 loss) and proposed PS loss. We could observe that the result with PS loss has a maximum growth in SSIM metrics (+0.008) without any additional parameters. It shows the proposed PS loss is good at restoring perceptually-faithful images.

**Skip connection.** The last component is the skip connection. We can observe the CMFNet with proposed MSC yields the best performance (29.16 dB), which also validates the effectiveness of our MSC design. In the process of our experiments, the gradient exploding problem occurs, so we decrease the initial learning rate from $2 \times 10^{-4}$ to $10^{-4}$. However, the network with skip connection in the last two rows can successfully be trained with the initial learning rate of $2 \times 10^{-4}$.

Table 4: Ablation study results of CMFNet. The last two columns are the average PSNR and SSIM of GoPro testing images.

| # of Branch | Attention block | Loss | SC | PSNR | SSIM |
|---|---|---|---|---|---|
| 1 | Channel attention (CA) | L1 loss | ✗ | 28.00 | 0.858 |
| 1 | Spatial attention (SA) | L1 loss | ✗ | 28.26 | 0.859 |
| 1 | Pixel attention (PA) | L1 loss | ✗ | 28.63 | 0.868 |
| 2 | CA+ SA | L1 loss | ✗ | 28.58 | 0.873 |
| 2 | CA+ PA | L1 loss | ✗ | 28.76 | 0.870 |
| 2 | SA+ PA | L1 loss | ✗ | 28.81 | 0.872 |
| 3 | CA+ PA+SA | L1 loss | ✗ | 28.88 | 0.873 |
| 3 | CA+ PA+SA | PS loss | ✗ | 28.90 | 0.881 |
| 3 | CA+ PA+SA | PS loss | ✔(ASC) | 29.08 | 0.886 |
| 3 | CA+ PA+SA | PS loss | ✔(MSC) | **29.16** | **0.887** |

## 5 CONCLUSION

In this paper, we proposed a general framework for image restoration which mimics the multi-path RGCs, called CMFNet. It could restore the degradation types, including blur, haze and rain, which usually occur on the images taking by the camera of self-driving. Besides, we proposed a novel loss function for general restoration tasks and the MSC to replace the traditional skip connection. Both of them are proved that they are effective in improving the restoration performance. Furthermore, our model achieves significant SSIM gain for four different datasets with different degradation types, which demonstrates the perceptually-faithful restored images. In the future, we are going to attempt more different restoration tasks, such as low-light image enhancement, streak rain removal, and the defocus blur. Additionally, we will also focus on balancing the trade off between the accuracy and computational efficiency.

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

## A APPENDIX

### A.1 DEBLUR VISUAL PERFORMANCE

Figs. 7, 8 show the visual deblur performances on the GoPro, HIDE datasets by our proposed CMFNet. Although our average PSNR and SSIM are not the best, it still has the competitive performance in deblurring.

### A.2 DEBLUR DISCUSSION

Fig. 9 reports our training loss and validation performance for deblur training process. We can see that the loss does not converge yet, and the validation score (PSNR) is also growing. We train our deblurring model for 10 days by one 1080Ti GPU, and just stop at 94 epochs. In the future, we will keep training till the loss is convergent in order to obtain the best performance for the deblurring task.

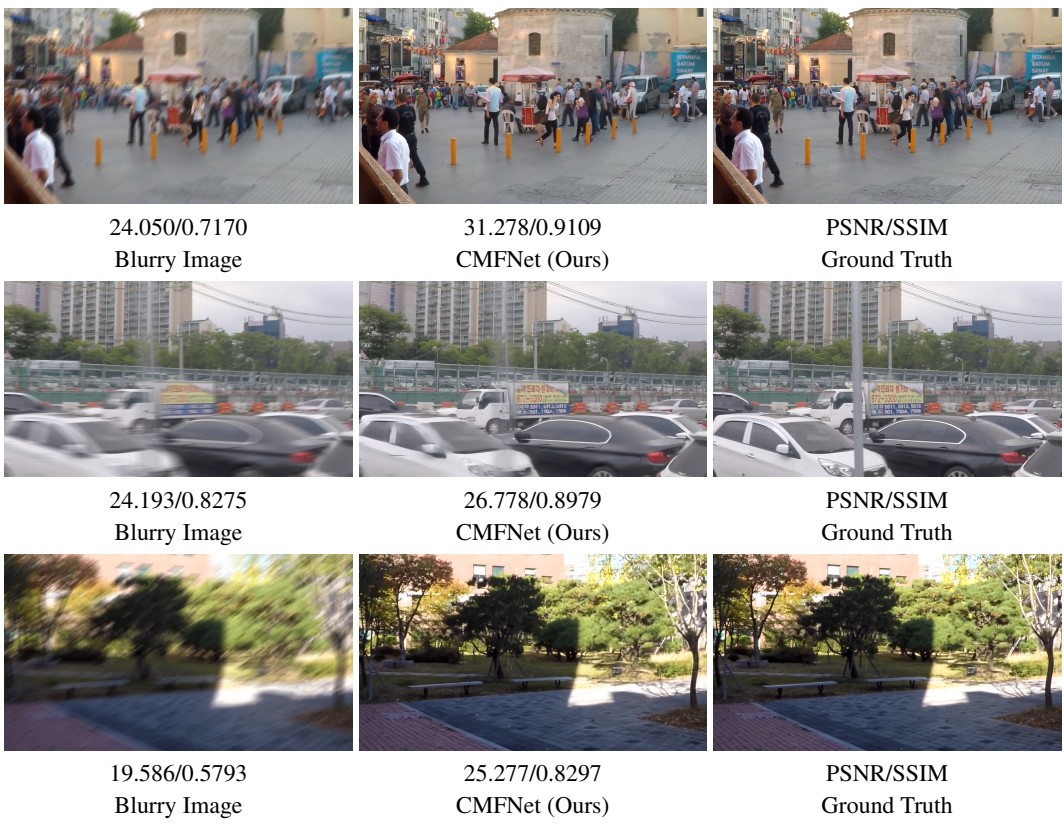

| 24.050/0.7170 | 31.278/0.9109 | PSNR/SSIM |
| Blurry Image | CMFNet (Ours) | Ground Truth |

| 24.193/0.8275 | 26.778/0.8979 | PSNR/SSIM |
| Blurry Image | CMFNet (Ours) | Ground Truth |

| 19.586/0.5793 | 25.277/0.8297 | PSNR/SSIM |
| Blurry Image | CMFNet (Ours) | Ground Truth |

Figure 7: Visual performances for image deblurring on the GoPro dataset.

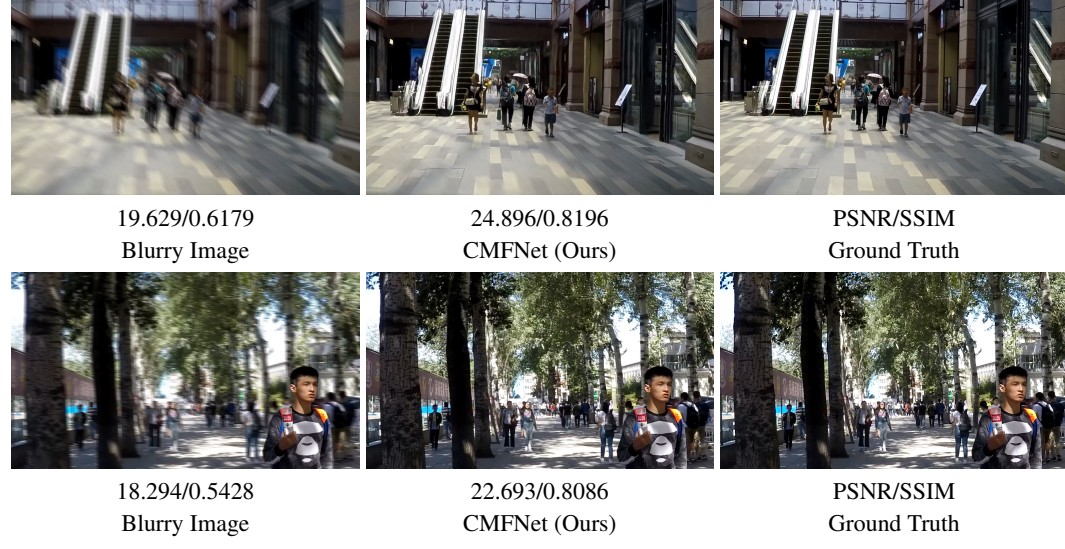

| 19.629/0.6179 | 24.896/0.8196 | PSNR/SSIM |
| Blurry Image | CMFNet (Ours) | Ground Truth |

| 18.294/0.5428 | 22.693/0.8086 | PSNR/SSIM |
| Blurry Image | CMFNet (Ours) | Ground Truth |

Figure 8: Visual performances for image deblurring on the HIDE dataset.

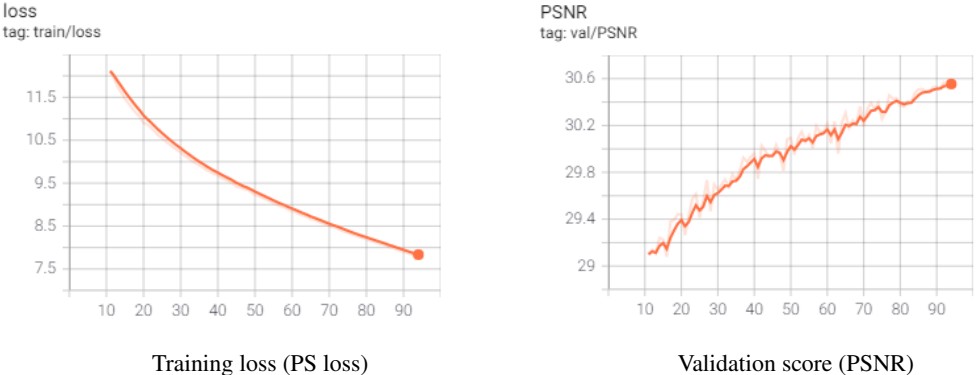

Training loss (PS loss)                    Validation score (PSNR)

Figure 9: Training loss and validation curve for deblurring.

