# OpenReview forum: "Compound Multi-branch Feature Fusion for Real Image Restoration"
_ICLR.cc/2022/Conference — ICLR 2022 Submitted_

### Official Review · Reviewer_sRLq · 2021-10-25

**Correctness:** 3
**Technical Novelty And Significance:** 2
**Empirical Novelty And Significance:** 2
**Recommendation:** 3
**Confidence:** 4

**Main Review:**

Strengths:
1. A compound multi-branch feature fusion architecture is proposed to simulate the complex human visual system for better restoration.
2. This paper tends to improve the generalization ability of a single restoration model for different degradation types.

Weaknesses:
1. Although the motivation to improve the model generalization ability is good, this paper does not clearly explain why the proposed architecture could actually do this. The authors state that they propose a multi-branch feature fusion network to simulate the human visual system, but the proposed architecture is by nature a simple combination of three existing attention blocks. And the authors relate these blocks with some types of cells in the human visual system. However, the so-called generalization of the proposed architecture is mainly derived from different training data for various tasks.
2. In Table 4, what does ASC denote? Please explain it in both the table caption and main text. To better understand the effect of the mixed skip connection, it would be better to show the values of a(theta) in the training process.
3. Since this paper states that their performance is closer to the human sense and they already include both PSNR and SSIM metrics in their loss function, it would be more convincing to use some other image quality assessment methods, e.g., NIQE [Mittal et al., Making a completely blind image quality analyzer. IEEE Signal Processing Letters 2012], LPIPS [Zhang et al., The unreasonable effectiveness of deep features as a perceptual metric. CVPR 2018], etc.
4. According to Section 3.1, each RAM in Figure 1 should have another input of the degraded image.
5. To verify the effectiveness of the proposed method, it would be better to compare with [Chen et al., Pre-Trained Image Processing Transformer, CVPR 2021], which presents an architecture that can handle various image processing tasks. It also evaluates on image deraining.
6. The statement that the supervised loss in SAM will limit the learning of network lacks justification.
7. Overall, the experimental results are not impressive and even not good.
For image dehazing, although the SSIM result is the best, the visual result by the proposed method is clearly not good compared to that by AECR-Net in Figure 5. There are distinct haze residuals in the image recovered by this method.
For image deraindrop, it is hard to see that the proposed method performs better than other state-of-the-art methods.
For image deblurring, the quantitative performance of the proposed approach is much worse than other methods. In Figures 7-8, it is necessary to include the corresponding results by other methods. Or it is not convincing to demonstrate the effectiveness of the proposed method.


**Summary Of The Paper:**

This paper proposes a multi-branch restoration model to handle multiple restoration tasks in a general framework, including image dehazing, deraindrop, and deblurring. Specifically, the authors develop a multi-branch architecture with three types of existing attention blocks to simulate the Retina Ganglion Cells in the Human Visual System. The mixed skip connection with a learnable parameter is proposed to integrate the information from difference branches.

**Summary Of The Review:**

My biggest concerns are the first and last comments as listed above in the weaknesses. This paper does not clearly explain why the proposed architecture can help improve the generalization ability for various tasks. In addition, the experimental results are not good enough to demonstrate the effectiveness. Therefore, I recommend this paper with rejection.

---

### Official Review · Reviewer_MkTj · 2021-11-02

**Correctness:** 2
**Technical Novelty And Significance:** 2
**Empirical Novelty And Significance:** 2
**Recommendation:** 3
**Confidence:** 4

**Main Review:**

[Strengths]

While most image restoration methods employ one or two attention types, this method combines 3 of them. The proposed loss function ($\mathcal{L}_{ps}$) seems to be more effective than L1 loss.

[Weaknesses]

1. Motivation and the contributions

The authors claim that the proposed method is inspired by the human visual system, the composition of retinal ganglion cells, however, I don’t agree.
The used attention modules are already present and have been widely studied in previous works and no such works claim the biological connection.
While the authors may claim that combining them is novel, I don’t find the connection between the proposed MSC module for aggregation and the biological systems.

The authors claim that the proposed M-cells network and the K-cells network each handle the luminance and the color information but there is no supporting proof or analysis. As the attention module operates on feature space, such an interpretation is hard to build. I believe this is a jump of logic without sufficient grounds.

2. Mixed Skip Connection

MSC module produces a weight $a$ and puts the weight on $I_C$, $I_P$, $I_S$ with $(1-a)/2$, $a$, $(1-a)/2$.
I can understand that the weights are designed to satisfy a sum-to-one constraint. However, I don’t understand why the weights for $I_C$ and$ I_S$ are set equal and why only the weight for $I_P$ differs from them.


3. Related Works

For image dehazing, He et al. (2010) proposed **dark channel prior**, not deep channel prior.

In image deraining section, I suggest the authors maintain a formal tone rather than using colloquial language.

4. References

On page 1 3rd paragraph, (Chen et al., 2016) is not an image restoration paper. It is an image semantic segmentation paper.

Nah et al., 2017a and 2017b are duplicated.

Please unify the bibliography format
(Waqas Zamir et al., 2021) -> (Zamir et al., 2021)

(Zhao et al., ECCV 2020) is not correct. It was published in ECCV 2020 **Workshop**.
https://link.springer.com/chapter/10.1007/978-3-030-67070-2_3

5. Task Generalization

In the abstract, the authors claim that the previous method lack generalization to multiple degradation types. However, the proposed method does not lead to task generalization. It seems that the authors trained the model independently to each task which could have been done by other methods, too. This is a misleading claim.

6. Experimental results

The ablation study shown in Table 4 is not fair.
By adding blocks, the models would have increased numbers of parameters. The results are not beyond expectation.

**Summary Of The Paper:**

This paper proposes an attention-based image restoration neural network architecture. The authors employ 3 types of attention methods in parallel: pixel attention, channel attention, and spatial attention. The outputs from the 3 attention-based UNet modules are aggregated by another module, mixed skip connection (MSC). The proposed method is applied to image dehazing, deraining, and deblurring. To train the model, the authors proposed a loss function using PSNR and SSIM.



**Summary Of The Review:**

Considering the strengths and weaknesses, the contributions are minor and the authors’ interpretation of the proposed method seems void.

---

### Official Review · Reviewer_WqhR · 2021-11-02

**Correctness:** 3
**Technical Novelty And Significance:** 2
**Empirical Novelty And Significance:** 3
**Recommendation:** 6
**Confidence:** 4

**Main Review:**

The paper is well structured, written succinctly and presents an interesting idea vaguely inspired by the human visual system. The drawn comparison between P-, M- and K- cells of retina ganglion and attention mechanisms is intriguing. The paper also includes insightful observations and thorough prior work overview. The proposed model shows impressive qualitative results of image restoration, significantly outperforming related works visually (Figure 5 and 6).

However, there are also a few concerns:
1. Quantitative evaluation of the image restoration performance is based only on two metrics, PSNR and SSIM. While on it’s own those metrics are valid and representative, the problem arises from the fact that PSNR and SSIM are also directly used in a loss function for the model (PS loss term, section 3.4). Essentially, during training the model is optimized on metrics that are used to evaluate the model.
That does not invalidate the proposed approach or loss function, but highlights that evaluation requires more thought. I strongly suggest adding additional independent image quality assessment metrics, for example MAE, MSE or at least modified SSIM metrics: MSSIM, FSIM, IW-SSIM.

2. Paper does not include any mention of important performance details, such as number of parameters, memory usage or training stability. For example, Table 4 (ablation study results) would strongly benefit from the first two, while section 3.4, describing novel loss function, would benefit from information about the training process.

3. Few basic model design decisions in the paper are not backed by anything other than intuition. One of the most noticeable examples is in the description of novel Residual Attention Module block, adapted from Supervised Attention Module by removing loss function:

    >  “We also remove the supervised loss between the output images from SAM and ground truth images, because we think it would limit the learning of network.”

    I believe additional argumentation or at least some empirical backing can improve the clarity and explain motivation behind such design decisions.


**Summary Of The Paper:**

The paper “Compound multi-branch feature fusion for real image restoration” proposes CMFNet, an image restoration model inspired by retina ganglion cells (RGC) of the human visual system. Namely, authors draw comparisons between parts of RGC and attention mechanisms: P-, M-, K-cells and pixel, channel, spatial attention respectively. CMFNet infers the attentions in parallel branches and fuses them by using proposed Residual Attention Modules and Mixed Skip Connection blocks to obtain a restored image. The model is trained by a novel loss function that integrates PSNR and SSIM image quality metrics. The experiments described in the paper show competitive results.

**Summary Of The Review:**

The paper proposes an intriguing model inspired by the human visual system and shows impressive qualitative results. At the same time, quantitative evaluation definitely requires more work. The paper would benefit from the addition of practical details about the model and more explanation about motivation behind some design decisions.

---

### Official Review · Reviewer_ttvV · 2021-11-02

**Correctness:** 2
**Technical Novelty And Significance:** 2
**Empirical Novelty And Significance:** 1
**Recommendation:** 1
**Confidence:** 5

**Main Review:**

## Main strengths:
1) The authors propose a network that can handle multiple image restoration tasks with comparable or better performance than some of recent methods that are directly designed for each of the considered tasks.
2) They also justify their design choices with inspirations from human visual system.

## Main Weaknesses:
3) The authors have discussed how different branches of the proposed architecture resemble RGCs. However, the discussions are in high level and do not quite make the link as strongly as they claim, for example "PAB and P-cells have one-to-one correspondence". There needs to be a more detailed justification of how each of the paired paths resemble each other. This might have been shown in previous experimental studies on the RGCs, providing some estimation of the function that each path calculates and how the chosen architectures for each path also to some extent calculate a similar function. The main thing that I can understand from the text is that among the three considered choices each one is correctly matched to the related paths in RGCs. But is there a reason or an empirical study that shows these are good design choices and many other commonly used encoder-decoder networks are not?
4) Using skip connections with learnable components to gather information from different parts of the network in general, cannot be claimed as a novel contribution. For example it is a common component of gradient-based neural architecture search methods. Refer to "Neural Architecture Search: A Survey" by Elsken et al. for a few examples.
5) The ablation study on "Number of branches" claims in the text that going from one to three branches improves the PSNR "from 28.00 dB
to 28.88 dB". However, table 4 shows that the best result obtained by only one branch is 28.63. So, the statement should be corrected. Overall, the improved performance added by using additional channels seems marginal to me. The ablation also does not consider the effect of added parameters when using more than one branch. It could be that a single branch of Pixel attention with equal number of parameters as the three branches together perform better than the proposed combination of branches. Another possible configuration that is missing is repeating the PA branch instead of using the other choices (with considerations to equalize the number of parameters). Unless such ablations are provided, I cannot be sure the proposed combination has any empirical advantage.
6) The authors have chosen to use a two years old version of the dehazing task (ntire'19) rather than the latest version. There are also missing SOTA papers from this year's CVPR and ICCV conferences with reported results on the chosen dataset such as "A Two-branch Neural Network for Non-homogeneous Dehazing via Ensemble Learning" by Yu et al. and "DW-GAN: A Discrete Wavelet Transform GAN for Non Homogeneous Dehazing" by Fu et al. which seem to be much better than the proposed method in terms of PSNR and SSIM. The authors should include those in their comparisons.
7) There are also missing SOTA papers from this year's CVPR and ICCV conferences with reported results on the deraindrop dataset such as "Spatially-Adaptive Image Restoration using Distortion-Guided Networks" by Purohit et al. and "Removing Raindrops and Rain Streaks in One Go" by Quan et al. which seem to be much better than the proposed method in terms of either SSIM or both PSNR and SSIM. The authors should include those in their comparisons.
8) For the task of deblurring, there are many published papers that have much better performance. A few examples from this year's CVPR and ICCV conferences are: "SDWNet: A Straight Dilated Network with Wavelet Transformation for image Deblurring" by Zou et al., "Perceptual Variousness Motion Deblurring with Light Global Context Refinement" by Li et al., "Training Weakly Supervised Video Frame Interpolation with Events" by Yu et al., "Bringing Events into Video Deblurring with Non-consecutively Blurry Frames" by Shang et al., and "Pyramid Architecture Search for Real-Time Image Deblurring" by Hu et al. The authors should include some of those in their comparisons.

#### Minor points:
9) The phrase in the abstract "which means they are lack of generalization" could be modified to "which points to their lack of generalization"
10) "I_C ∈ R^{H×W×3} denotes the output image obtained by passing the degraded image through 3 × 3 convolution." What I understand from this statement is that there should be a connection from the input degraded image to a 3x3conv module and then the output would be I_C. However, Fig.1 shows that I_C is output of the RAM module on the top which itself is getting its input from the previous encoder-decoder module. It seems like either the statement or the figure should be corrected.
11) Regarding the L_PS portion of the loss, the authors claim that the formulation results in avoiding to consider a ratio is not necessarily true. Just because avoiding to consider a ratio has worked, does not mean considering one and experimenting with different values of it would not impact the results to be better or worse. such a scale ratio could be multiplied by the PSNR as an example. Note that the loss has another component and this ratio should be empirically studied in parallel to the (\epsilon).

**Summary Of The Paper:**

The paper addresses the issue of multi-task image restoration. Inspired by the human visual system, the authors propose a multi-branch encoder-decoder architecture to address three tasks, dehazing, deraindrop, and deblurring, which coincide together, for example in autonomous driving scenes. They also propose to fuse the information using skip connections with a learnable parameter controlling the mixture ratios. Their results are comparable in all of the three tasks to some recent methods targeted for each individual task in terms of SSIM.

**Summary Of The Review:**

The paper provides a nice intuition on how to simulate the human visual system in terms of designing a generic architecture that can work for multiple tasks. However, the mentioned points of weakness substantially impact the quality of the work. These include the proposed design (3), claimed contributions (4), empirical approach to ablations (5), and the fact that results are far worse than the state of the art (6,7,8). I have provided the details above with feedback on how to address each point). To address the mentioned concerns and achieve at least somewhat comparable performance to the SOTA, the paper needs both much further experiments and major revisions. Therefore, I cannot recommend the current version for the conference. However, this should not stop the authors from improving on their current work and hopefully submit to a future conference when it is ready.

---

### Official Review · Reviewer_eNZ2 · 2021-11-04

**Correctness:** 2
**Technical Novelty And Significance:** 2
**Empirical Novelty And Significance:** 2
**Recommendation:** 3
**Confidence:** 5

**Details Of Ethics Concerns:**

No concerns.

**Main Review:**

Strengths
1. The idea to handle multiple restoration tasks in one model is meaningful and reasonable.
2. The way of using different kind of attention module in parallel branches is interesting and new.
3. The paper writing is clear and easy to follow.


Weaknesses
1. The idea of using parallel branches for each kind of attention module to mimic HVS seems interesting.
    However, the connection between P/M/K-cell and the 3 kinds of attention module is not very convincing.
    The explanation in Sec 3.2 gives abstract analysis. However, no proof and visualizations are given.
    It would be better to show and visualize the actual feature maps and the effect of different branches.

2. The ablation study part is not very fair.
     In Sec 4.6, the experiment 'number of branch' compares model with different trainable parameters.
     The three branch model is more complex than one branch model, which may be the reason for performance gain.

3. My main concern is about the novelty part. The whole network seems a combination of existing modules, without a clear clue why they are suitable for restoration tasks?
    What are the specific characteristics for restoration tasks? Why the designed network is suitable? And how to visualize the benefits of intermediate results?

4. In Sec 4.2 and 4.3, I wonder does the model train on 3 datasets simoutanously? Or the authors train 3 model for different tasks each?
    Actually, there already exists method to handle real degradation simoutanously. Please at least cite or compare:
[1] Real-ESRGAN: Training Real-World Blind Super-Resolution with Pure Synthetic Data, ICCVW 2021.
[2] ESRGAN: Enhanced super-resolution generative adversarial networks, ECCVW 2021.

5. The authors mentioned many times that they can handle real images.
    However, they only use existing dataset for training and testing.
    Even if the proposed method can really handle real degradation, the contribution comes from previous works.
    The authors do not explain why their model can do this.

6. Several typos and mistakes. For example, in Eq (2), the authors say X denotes degradation images. Is X denoted for output images?

7. Some of the statements about previous methods are not accurate.
    a. In Sec 2 'Image Dehazing', He et al.'s work is a very famous and foundamental work in that area. And it should be 'dark channel prior'.

    b. In Sec 2 'Image Deblurring', they said those methods are `generally tested on synthesized images`. However, that is not correct.
    To my knowledge,
    [1] Two-Phase Kernel Estimation for Robust Motion Deblurring, ECCV 2010
    [2] Removing camera shake from a single image, SIGGRAPH 2006
    [3] Fast Motion Deblurring, SIGGRAPH 2009
    [4] Non-uniform Deblurring for Shaken Images, CVPR 2010
   They are all very famous early works and they all test on real captured photos.

   On the contrary, this paper only show results on testing dataset, and do not compare photos that from a different source from training dataset.




**Summary Of The Paper:**

This paper proposes a new model CMFNet to handle multiple image restoration tasks. More specifically, they introduce a new skip connection scheme, a new loss function and demonstrate comperitive results on 3 tasks: deblurring, deraining and dehazing.

**Summary Of The Review:**

This paper proposes a new model CMFNet to handle multiple image restoration tasks. The results seem promising on several existing datasets. However, the proposed framework and modules seems similar to widely used techniques. And there is no clear and solid explanations for the design choice.

---

### Decision · Program_Chairs · 2022-01-20

**Decision:**

Reject

**Comment:**

All reviewers have substantial concerns regarding this work including novelty and experimental validation. The authors do not provide a rebuttal for the raised concerns. As such, the area chair agrees with the reviewers and does not recommend it be accepted at this conference.